# The Estimation of Chemical Properties of Pepper Treated with Natural Fertilizers Based on Image Texture Parameters

**DOI:** 10.3390/foods12112123

**Published:** 2023-05-24

**Authors:** Ewa Ropelewska, Justyna Szwejda-Grzybowska

**Affiliations:** Fruit and Vegetable Storage and Processing Department, The National Institute of Horticultural Research, Konstytucji 3 Maja 1/3, 96-100 Skierniewice, Poland; justyna.grzybowska@inhort.pl

**Keywords:** red pepper, yellow pepper, α-carotene, β-carotene, total carotenoids, total sugars, image processing, regression

## Abstract

The cultivar and fertilization can affect the physicochemical properties of pepper fruit. This study aimed at estimating the content of α-carotene, β-carotene, total carotenoids, and the total sugars of unfertilized pepper and samples treated with natural fertilizers based on texture parameters determined using image analysis. Pearson’s correlation coefficients, scatter plots, regression equations, and coefficients of determination were determined. For red pepper Sprinter F_1_, the correlation coefficient (*R*) reached 0.9999 for a texture from color channel B and −0.9999 for a texture from channel *Y* for the content of α-carotene, −0.9998 (channel *a*) for β-carotene, 0.9999 (channel *a*) and −0.9999 (channel *L*) for total carotenoids, as well as 0.9998 (channel *R*) and −0.9998 (channel *a*) for total sugars. The image textures of yellow pepper Devito F_1_ were correlated with the content of total carotenoids and total sugars with the correlation coefficient reaching −0.9993 (channel *b*) and 0.9999 (channel *Y*), respectively. The coefficient of determination (R^2^) of up to 0.9999 for α-carotene content and the texture from color channel *Y* for pepper Sprinter F_1_ and 0.9998 for total sugars and the texture from color channel *Y* for pepper Devito F_1_ were found. Furthermore, very high coefficients of correlation and determination, as well as successful regression equations regardless of the cultivar were determined.

## 1. Introduction

Nowadays, the consequences associated with conventional agricultural production are observed, such as the disruption of natural diversity, soil and water pollution by fertilizers and plant protection products, reduced biomass production, and an increase in diseases. Therefore, there is an increasing trend in agricultural production based on the agroecological concept. With the ever-increasing nutritional awareness of society, the group of consumers looking for organically produced food is growing, and thus the demand for products with high health-promoting value and free of pesticides is increasing. The use of chemical plant protection products is an easy way of combating organisms that are harmful to plants, but it carries a risk to the health of people, animals, and the environment. More and more modern preparations of natural origin for use in plant cultivation are appearing on the market, which have a beneficial effect on plant-life processes and, at the same time, can influence the composition and content of bioactive and nutritional compounds and protect plants from diseases and pests [1,2]. 

Sweet pepper (*Capsicum annuum* L.), otherwise known as the annual pepper plant, is classified as a plant of the Solanaceae family. About 39 pepper species have been identified, including five domesticated species, *C. annuum*, *C. baccatum*, *C. chinense*, *C. frutescens*, and *C. pubescens* [3]. It is a vegetable with low energy value and low index and glycemic load. It is one of the vegetables characterized by high health-promoting compounds, carotenoids (especially beta-carotene, α-carotene, and lutein); phenolic compounds; and vitamins, such as ascorbic acid, vitamins (A, E, and K). It is also a good source of folic acid, minerals (potassium, manganese, iron, and magnesium), and dietary fiber [4,5,6]. In previous studies, a 100 g serving of peppers was found to cover 100% of the recommended daily intake of vitamin C, 5–10% of tocopherol, and 5–10% of provitamin A [6].

Nutritional and health-promoting compounds contained in pepper fruits play an important role in the prevention of cardiovascular diseases and cancer [7,8,9]. Among these compounds, the carotenoids—pigments present in the seed coat of pepper fruits—are of great importance. Scientists are increasingly paying attention to the health-promoting value of carotenoids and their role in disease prevention. Among other things, these compounds are part of the body’s defense system against free radicals, an excess of which accelerates cellular aging processes. These processes can become the cause of the development of chronic diseases, such as cancer and cardiovascular diseases. As the selected are precursors of vitamin A, they also counteract diseases associated with vitamin A deficiency. Vitamin A is responsible for vision and is involved in processes in the retina. It also plays a very important role in the proliferation of bone tissue and epithelial cells, especially of the oral mucosa, gastrointestinal tract, urinary tract, respiratory tract, as well as the organ of vision. In plants, carotenoids are pigments that play an important role in protecting plants from photo-oxidative effects. They are effective antioxidants and participate in scavenging singlet oxygen and superoxide radicals [10].

Among the compounds that largely determine the nutritional value of pepper fruits are the sugars that give them their sweet taste. Their content is genetically determined and can be modified by agrotechnical and climatic conditions [11,12,13,14]. The sugar content also depends on the ripeness of the pepper fruit [13,15,16]. These compounds mainly accumulate in the pericarp of peppers and their content may vary from variety to variety [17]. In addition, sugars, together with specific volatile compounds, can affect the overall perception of taste and are vitamin C stabilizers [18]. Vegetables and fruits that have a high vitamin C content in their composition have been proven to have high levels of sugars [13,19]. 

The pepper fruit cultivars differ in color, size, shape, and flavor. The color and shape of pepper fruit at the stage of consumer maturity depend on the presence of a gene system. Pepper fruits synthesize and accumulate a variety of carotenoids in tissues with a wide range of content. Even within the same cultivar, the composition and number of carotenoids can vary considerably depending on maturity stage, growing region, and growing conditions [20,21,22,23]. The first level of regulation of carotenoid biosynthesis in pepper fruit is the transcriptional control of gene biosynthesis. It is the main determinant of carotenoid production in pepper during fruit ripening in response to developmental cues [24,25]. Various studies confirm that capsanthin, β-carotene, and β-cryptoxanthin are the carotenoids responsible for the red color of peppers [26,27]. In contrast, yellow pepper varieties contain lutein, violaxanthin, antheraxanthin, and zeaxanthin. The orange color of pepper fruits is determined by the presence of α- and β-carotene, β-cryptoxanthin, zeaxanthin, and lutein. Brown varieties mainly contain red carotenoid pigments, such as red pepper fruits [26]. The shape of the pepper fruit is also an important varietal characteristic, ranging from very long and narrow, through spherical to fruit in the form of a short or elongated prism. The wall thickness of the pepper fruit can be 5–12 mm depending on the variety and growth conditions. The average weight of pepper fruit in large-flowered sweet varieties reaches 250–350 g, and often more [13]. Flesh and skin texture, juiciness and postharvest shelf life are determined by both genetic traits and growing conditions [28].

As mentioned above, the physicochemical properties depend on the pepper cultivar and can be affected by fertilization. Research involving image analysis allows for an objective assessment of the quality of food products with the reduction in the time, costs, and labor consumption of analyzes [29]. Due to image processing, the textural, color, and morphological parameters in the images can be computed and then interpreted [30]. The image texture parameters can be considered a function of the spatial variation of the brightness pixel intensity and carry information about the structure of the samples, whereas analyses of image textures provide insights into product quality. Image textures can be analyzed using specialized software packages, for example, MaZda, which enables quantitative analysis, including texture parameters computation, as well as texture selection and extraction, data classification and visualization, and image segmentation tools. MaZda provides options for the transformation of color images to grayscale and conversions to individual color channels (components) [31,32,33]. Several channels are most commonly used. There are color channels *R* (red), *G* (green), and *B* (blue) from the RGB color space; *L* (lightness component from black to white), *a* (red for positive and green for negative values), and *b* (yellow for positive and blue for negative values) from the Lab color space; and *X* (component with color information), *Y* (lightness), and *Z* (component with color information) from the XYZ color space [29,34].

The objective of this study was to estimate the chemical properties of peppers belonging to two cultivars treated with natural fertilizers in a fast and inexpensive manner based on image texture parameters. A great novelty of the present study is related to the estimation of α-carotene, β-carotene, total carotenoids, and total sugars of untreated red pepper Sprinter F_1_ and yellow pepper Devito F_1_ and samples subjected to different organic fertilization, such as NaturalCrop^®^ SL, Bio-algeen S90, and nettle fertilizer. The innovative regression equations including features selected from a set of 1629 image textures from nine color channels were determined. The developed procedure could enable the selection of fruit with the desired properties based on the image features without the need to perform more labor-intensive and expensive measurements.

## 2. Materials and Methods

### 2.1. Experimental Design

The experimental material consisted of pepper growing in greenhouses at the Laboratory of Cultivation of Vegetable and Edible Mushrooms of the National Institute of Horticultural Research in Skierniewice in Poland. Peppers belonged to two cultivars of Sprinter F_1_ characterized by red fruit and Devito F_1_ with yellow fruit. The experiment was carried out between March and October 2021. During the experiment, daily climatological data were collected inside the greenhouse, using a thermo-hygrometer. Organic pepper seeds were used and potted seedlings were planted in the soil in an ecological greenhouse under the principles of organic farming. The experiment was set up in the soil with a pH of 6.0. Before planting, the soil was fertilized with organic plant compost at a dose of 300 kg per 100 m^2^, obtaining the soil fertility at the level (mg dm^−3^) of N-230, P-310, K-400, and Mg-350. The seedlings were planted on April 20, in a row-row system, at a spacing of 35 cm × 60 cm (3.6 plants per 1 m^2^). Plants were grown in a four-shoot system. Pepper fertigation was carried out with the use of a drip system and fertilizers approved for organic cultivation depending on the current humidity and nutritional requirements of the pepper in particular growth stages [35]. The fruit was harvested at the stage of colorful ripeness.

### 2.2. Preparations

In research natural preparation and fertilizers approved for organic farming was performed such as: nettle fertilizer (NE/427/2018) (as a substance with biocidal and nutritional effect) based on N-NO_3_ about 0.5–0.7 mg·L^−1^, N-NH4 >100 mg·L^−1^, P about 20 mg·L^−1^, K-about 650 mg·L^−1^, F—0.2 mg·L^−1^, pH 6.3–6.7 (20% solution was used); Bio-algeen S90 (organic fertilizer) containing 90 groups of chemicals compounds, including amino acids; vitamins; alginic acid and other unexplored active components of marine algae; N—0.02%; P—0.006%; K—0.096%; Ca—0.31%; MG—0.021% and B—16 mg·kg^−1^; Fe—6.3 mg·kg^−1^; Cu—0.2 mg·kg^−1^; Mn—0.6 mg·kg^−1^; Zn −1.0 mg·kg^−1^ (0.3% solution was used); NaturalCrop^®^ SL (growth and development stimulator)—enzymatic concentrate of peptides; and 16 L-amino acids most important for plants (GLY, GLU, HIS, LYS) of plant origin >50% (605 g·L^−1^), including free amino acids 0.2% (242 g·L^−1^), amino nitrogen 9% (109 g·L^−1^), organic carbon 24.5% (296.5 g·L^−1^) (0.3% solution was used), and the control (unfertilized plots).

### 2.3. Image Acquisition and Processing

The image acquisition was performed using an Epson Perfection V600 flatbed scanner. The scanner was placed in a black box to obtain a black background of images. The images of pepper slices (cross-sections) with a thickness of 10 mm obtained using a slicing machine were considered. In the case of each cultivar and treatment, images of 50 slices were acquired, as follows: − Fifty slices of a control sample of Sprinter F_1_ pepper,− Fifty slices of Sprinter F_1_ pepper treated with NaturalCrop^®^ SL, − Fifty slices of Sprinter F_1_ pepper treated with Bio-algeen S90,− Fifty slices of Sprinter F_1_ pepper treated with nettle fertilizer,− Fifty slices of a control sample of Devito F_1_ pepper,− Fifty slices of Devito F_1_ pepper treated with NaturalCrop^®^ SL, − Fifty slices of Devito F_1_ pepper treated with Bio-algeen S90,− Fifty slices of Devito F_1_ pepper treated with nettle fertilizer.

The acquired images were saved in a TIFF format at 800 dpi resolution. Before processing in the Mazda software (Łódź University of Technology, Institute of Electronics, Łódź, Poland) [31,32,33], the pepper slice images were converted to a BMP format. Then, texture parameter extraction was carried out for each of the color channels *L*, *a*, *b*, *R*, *G*, *B*, *X*, *Y*, and *Z*. The sample images are presented in Figure 1. The image segmentation into pepper slices and the background was performed manually based on the pixel brightness intensity. Each pepper slice (the area of flesh without skin) was considered as a region of interest (ROI). For each ROI, in total 1629 image textures, including 181 textures for each color channel were computed. 

### 2.4. Chemical Properties

The samples for chemical analysis were prepared from 10 pepper fruits from each repetition. Pepper fruits were cut into quarters, frozen in liquid nitrogen, crushed in a Blixer homogenizer and stored at −20 °C. 

#### 2.4.1. HPLC Analysis of Sugars

An HPLC analysis of total sugars (sucrose, glucose, and fructose) was determined by high-performance liquid chromatography (Agilent 1200 HPLC system, equipped with a differential refractometric detector), using Aminex HPX-87C (300 mm × 7.5 mm) with a precolumn according to European Standard EN 12630. The isocratic flow was 0.6 mL·min^−1^, column temperature was 80 °C, and mobile phase-edetate calcium disodium (Ca-EDTA). The samples were dissolved in redistilled water, homogenized, and purified on a Waters SepPak PLUS C18 filter. The sugars were quantified by calibration curve for sucrose, glucose and fructose, and the results were expressed as mg 100 g^−1^ fresh mass (f.m.). 

#### 2.4.2. Carotenoid Extraction 

Carotenoid content was determined by the method according to Bohoyo-Gil et al. [36]. The sample was homogenized in the extraction solution (hexane:acetone 6:4) and filtered through a Büchner funnel under reduced pressure. Next, the extract was transferred to a separating funnel and shaken with the addition of water. After phase separation, the water–acetone phase was discarded. The acetone rinsing operation was repeated until the lower phase was free of acetone and the upper hexane phase containing carotenoids was filtered through a filter paper containing anhydrous sodium sulphate into an evaporation flask. Hexane was evaporated to dryness in a vacuum evaporator at 40 °C, the dry residue was quantitatively transferred to a 25 mL flask with a solution of acetonitrile:methanol:ethyl acetate 55:25:20 + 0.1% BHT + 1 mL TEA and 4 mL of hexane. The flask extract was filtered with a 45 µm PTFE filter into an amber bottle and analyzed by HPLC. 

#### 2.4.3. HPLC Analysis of Carotenoids 

An HPLC analysis of total carotenoids in pepper samples was determined by high-performance liquid chromatography (HPLC), using a Kinetex C-18 column (250 mm × 4.6 mm; 5 µm) on an Agilent 1200 HPLC system equipped with a DAD detector. The elution conditions were as follows, 0.7 mL min^−1^, temperature 28 °C, wavelength 450 nm, and 472 nm; in the mobile phase, acetonitrile (A), methanol + 1 mL TEA + 1 g BHT (B), ethyl acetate (C); and in gradient flow, 0–10 min, 95% A, 2% B, 3% C; 10–25 min, 55% A, 20% B, 25% C; 25–35 min, 95% A, 2% B, and 3% C. The calculations were quantified by calibration with the standards of β-carotene and α-carotene (Sigma-Aldrich, Darmstadt, Germany). The carotenoid content was expressed in mg 100 g^−1^ f.m.

### 2.5. Statistical Analysis

The linear relationships between the chemical properties and image textures of pepper samples were determined using STATISTICA 13.1 (Dell Inc., Tulsa, OK, USA, StatSoft Polska, Kraków, Poland). The analysis was performed separately for samples of Sprinter F_1_ and Devito F_1_ and then for a dataset combining both cultivars Sprinter F_1_ and Devito F_1_. In the case of Sprinter F_1_ pepper, among the chemical properties, α-carotene, β-carotene, total carotenoids, and total sugars were considered. For Devito F_1_, α-carotene was not detected and the differences in the content of β-carotene were very small. Therefore, the analyses for Devito F_1_ and a dataset combining Sprinter F_1_ and Devito F_1_ were performed only for total carotenoids and total sugars. In the first step, Pearson’s correlation coefficients (R) were determined at a significance level of *p* < 0.05, and scatter plots were created. In the case of image textures from color channels *L*, *a*, *b*, *R*, *G*, *B*, *X*, *Y*, and *Z* for which the correlations occurred, the highest value of R for positive correlation and the highest value of R for negative correlation were selected to be presented in this paper. Then, regression equations for the estimation of chemical properties of pepper untreated and treated with natural fertilizers using image texture parameters were determined and coefficients of determination (R^2^) were computed. 

## 3. Results and Discussion

### 3.1. Relationship between Chemical Properties and Image Texture Parameters of Red Pepper Sprinter F_1_

In the case of red pepper Sprinter F_1_, correlations between α-carotene, β-carotene, total carotenoids, and total sugars with selected image textures were observed (Table 1). Very strong positive and negative correlations were determined for each of the considered organic chemical compounds reaching 0.9999 and −0.9999 for α-carotene and total carotenoids. In the case of α-carotene, statistically significant correlation coefficients were found for textures of images in color channels *a*, *R*, *G*, *B*, *Y*, and *Z*. The correlation coefficient (R) of 0.9999 was found for one image texture from color channel *B* and −0.9999 for a texture from channel *Y*. None of the image textures from channels *L*, *b*, and *X* correlated with α-carotene. Therefore, these channels turned out to be useless for the estimation of α-carotene content. For total carotenoids, both positive and negative relationships with image textures from all color channels *L*, *a*, *b*, *R*, *G*, *B*, *X*, *Y*, and *Z* were observed, and the values of R equal to 0.9999 and −0.9999 were determined for color channels *a* and *L*, respectively. In the case of β-carotene, the positive correlations with image textures from color channels *a*, *R*, *G*, *B*, *Y*, and *Z* were observed, reaching 0.9994 for a texture from channel *B*. Whereas negative correlations with textures from color channels *a*, *R*, *G*, *B*, *X*, *Y*, and *Z* were observed, and the highest value was −0.9998 for a texture from channel *a*. It was found that the content of total sugars was positively correlated only with textures of images in color channels *R*, *G*, and *X* with the coefficient of up to 0.9998 in the case of color channel *R*. Negative correlations with textures from color channels *L*, *a*, *R*, *G*, *X*, *Y*, and *Z* reached the value of −0.9998 for channel *a*.

The scatter plots for α-carotene, β-carotene, total carotenoids, total sugars with selected image texture parameters, for which the highest values coefficient R for positive and negative correlations were determined, confirmed the strong relationship between chemical properties and image textures of control red pepper Sprinter F_1_ and samples treated with natural fertilizers (Figure 2). The positive relationships between α-carotene and the texture from color channel *B*, β-carotene and the texture from color channel *B*, total carotenoids, and the texture from color channel *a*, as well as total sugars and the texture from color channel *R*, are presented. The strongest negative correlations between α-carotene with the texture from color channel *Y*, β-carotene with the texture from color channel *a*, total carotenoids with the texture from color channel *L*, and total sugars with the texture from color channel *a* are shown.

### 3.2. Relationship between Chemical Properties and Image Texture Parameters of Yellow Pepper Devito F_1_

The texture parameters of flesh images of yellow pepper Devito F_1_ were correlated with the content of total carotenoids and total sugars (Table 2). Positive correlations were found between total carotenoids and image textures from color channels *L*, *a*, *b*, *B*, *X*, and *Y* with the value of correlation coefficient reaching 0.9899 for the texture from channel *X*, as well as total sugars and image textures from color channels *L*, *a*, *b*, *G*, *B*, *X*, and *Y* reaching 0.9999 for the texture from channel *Y*; while the content of total carotenoids was negatively correlated with image textures from color channels *L*, *b*, *R*, *B*, and *X* with a correlation coefficient of up to −0.9993 for the texture from channel *b*. The negative correlations between the content of total sugars and image textures from channels *L*, *a*, *G*, *B*, *X*, and *Y* reaching −0.9971 for the texture from channel *G* were determined. 

The relationships between total carotenoids and total sugars with selected Image texture parameters of control and fertilized yellow pepper Devito F_1_ are presented in Figure 3 in the form of scatter plots. The graphs for textures characterized by the strongest positive and negative correlations, such as textures from color channels *X* and *b* for total carotenoids, and textures from color channels *Y* and *G* for total sugars are shown.

### 3.3. Relationship between Combined Chemical Properties and Image Texture Parameters of Red Pepper Sprinter F_1_ and Yellow Pepper Devito F_1_

In the next step of the analysis, correlations between a set of combined image texture parameters and chemical properties of red pepper Sprinter F_1_ and yellow pepper Devito F_1_ belonging to the unfertilized group and samples treated with NaturalCrop^®^, Bio-algeen S90, and nettle fertilizer were determined (Table 3). The content of total carotenoids was positively and negatively correlated with image texture parameters from color channels *L*, *a*, *b*, *B*, *G*, *X*, *Y*, and *Z*. The highest positive and negative correlation coefficients were 0.9956 and −0.9935 for textures from color channel *Y*. In the case of total sugars, the positive correlations with image textures from channels *L*, *a*, *b*, *G*, *X*, and *Y* were observed and the correlation coefficient reached 0.9768 for the texture from channel *Y*; whereas the negative correlations with image textures from channels *L*, *a*, *b*, *G*, *B*, *X*, and *Y* with the highest value of −0.9583 for the texture from color channel *G* were determined.

The scatter plots for total carotenoids and total sugars confirmed the strong relationships with selected image texture parameters of red pepper Sprinter F_1_ and yellow pepper Devito F_1_ belonging to control and samples treated with NaturalCrop^®^, Bio-algeen S90, and nettle fertilizer (Figure 4). The strongest positive and negative correlations between total carotenoids with textures from color channel *Y* and total sugars with textures from color channels *Y* and *G* are presented. 

Additionally, the regression equations for estimating chemical properties of unfertilized and fertilized pepper samples, as well as coefficients of determination (R^2^) separately for red pepper Sprinter F_1_, yellow pepper Devito F_1_, and a set combining parameters for Sprinter F_1_ and Devito F_1_ were determined (Table 4). The higher R^2^, the better the data fit the regression model. Image textures whose values were the most different between groups of peppers were the most suitable for building regression models. In the case of each chemical parameter, one regression equation for a texture for which the strongest positive correlation and one equation for the strongest negative correlation were set up. In the case of pepper Sprinter F_1_, all values of R^2^ were higher than 0.9980, reaching 0.9999 for α-carotene content and the texture YS4RVLngREmph from color channel *Y*. It indicated that the texture YS4RVLngREmph most differentiated all groups of pepper Sprinter F_1_, such as control and samples treated with NaturalCrop^®^ SL, Bio-algeen S90, and nettle fertilizer. Pepper Devito F_1_ was also characterized by very high R^2^ of up to 0.9998 for the content of total sugars and the texture YS5SH1Entropy from color channel *Y*. It meant that the control sample of pepper Devito F_1_, and samples treated with three different natural preparations were the most different in terms of the texture YS5SH1Entropy that was most useful for developing the regression model; whereas the values of R^2^ reached 0.9912 for total carotenoids and the texture YS5SH5Contrast from color channel *Y* for a set combining parameters for both cultivars. This confirmed that the textures of color channel *Y* of images are the most successful for estimating chemical properties of untreated peppers and samples treated with natural fertilizers.

The performed studies revealed strong correlations between the chemical properties and selected image texture parameters of red and yellow pepper samples. Very correct regression equations turn out to be useful for the estimation of the content of carotenoids and sugars based on pepper image parameters. Additionally, available literature reported the relationships between image or spectral features and chemical characteristics. In our previous research, the correlations between lycopene content and image textures of tomato fruit were determined with the correlation coefficient (R) reaching −0.99 for selected texture parameters of images in channels *G*, *b*, and *Y* and coefficients of determination (R^2^) of 0.99 for image texture from channel *G* [37]. Pace et al. [38] observed relationships of the visual appearance and color parameters of images of fresh-cut nectarines with R^2^ of up to 0.7622. The sugar content of potatoes was predicted based on hyperspectral imaging data by Rady et al. [39] with R reaching 0.97 for glucose. The sugar content was also estimated for apples with a high R parameter (0.8861) using multispectral imaging [40] and wine grapes with R^2^ > 0.8 based on visible and near infrared and the short-wave infrared point spectroscopic data [41]. Furthermore, the prediction of soluble solid content of apples with R > 0.8 was possible based on features extracted from laser light backscattering images [42]. Our results presented in this paper were a significantly novel compared to the available literature, and included the determination of correlations and models for predicting the content of chemical properties based on selected parameters from a very large set of over 1600 textures from nine color channels. It allowed for the selection of textures providing correlation coefficients of up to 0.9999 and −0.9999, as well as very high coefficients of determination reaching 0.9999. The successful results may prompt the use of the developed procedure for other species and varieties of fruit and vegetables and the testing of the approach for other chemical parameters. 

## 4. Conclusions

The correlations between chemical properties and image textures from different color channels of red and yellow peppers untreated and treated with natural fertilizers were determined. In the case of red pepper Sprinter F_1_, statistically significant correlation coefficients were found between α-carotene, β-carotene, total carotenoids, and total sugars with selected image texture parameters. Yellow pepper Devito F_1_, as well as a combined dataset for both cultivars were characterized by the correlations between total carotenoids and total sugars with selected textures. Furthermore, the high coefficients of determination confirmed the possibility of estimation of chemical properties based on image texture parameters from different color channels for individual cultivars, as well as regardless of the cultivar. The highest values of R^2^ were for regression equations set up based on selected textures from images in color channel *Y*. 

Due to the achievement of promising results, research aimed at estimating chemical properties based on image features can be continued and further research directions can be set. Further research may include more pepper cultivars, as well as other species of fruit and vegetables. Furthermore, more natural fertilizers can be used to provide more groups in an experimental design. The developed procedure can be of great practical importance to estimate the chemical properties of samples treated with various fertilizers based on selected image texture parameters. 

## Figures and Tables

**Figure 1 foods-12-02123-f001:**
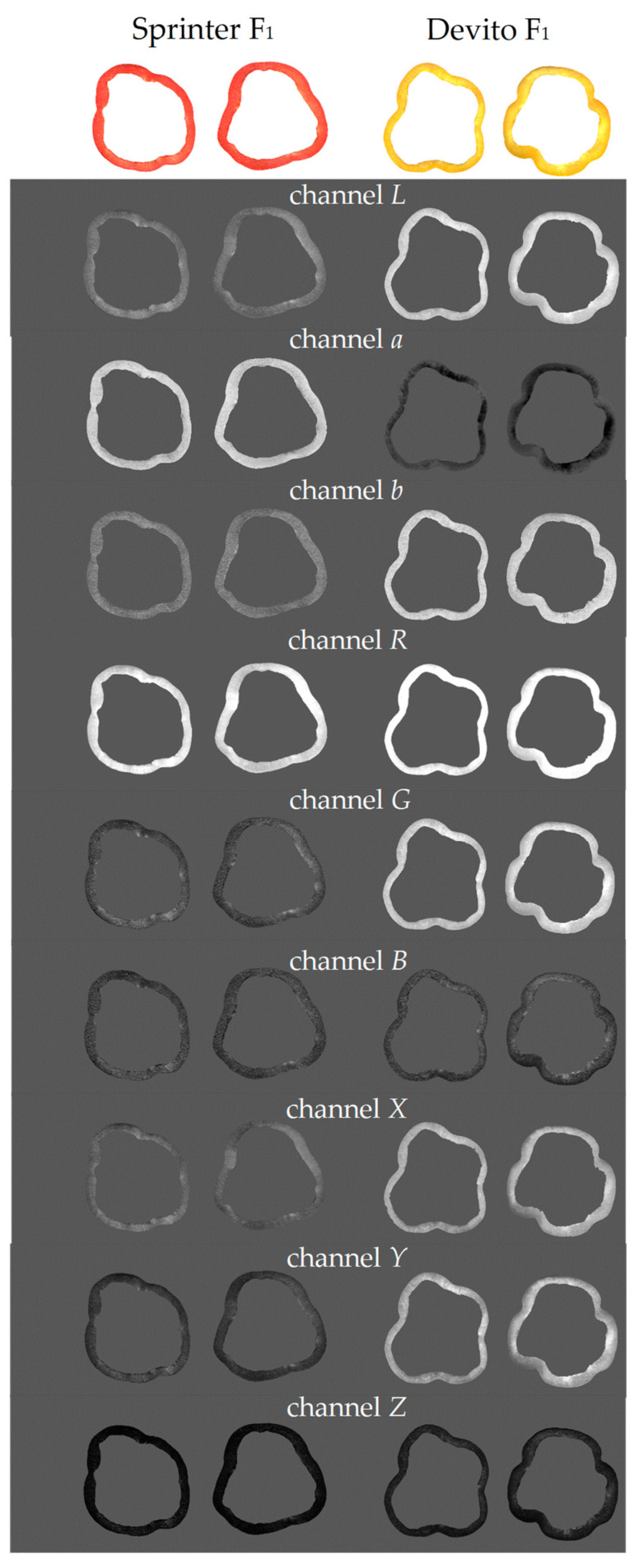
Original color images of pepper slices and images in different color channels.

**Figure 2 foods-12-02123-f002:**
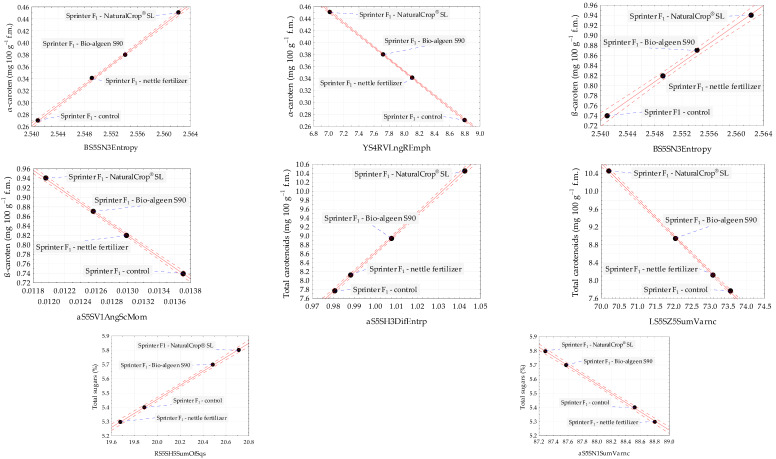
Scatter plots for α-carotene, β-carotene, total carotenoids, and total sugars with selected image texture parameters of control and treated with NaturalCrop^®^, Bio-algeen S90, and nettle fertilizer red pepper Sprinter F_1_ samples. Black dot—mean value; blue dashed line—line connecting the sample name to the mean value; red dashed line—confidence interval (95%); and solid red line—regression line.

**Figure 3 foods-12-02123-f003:**
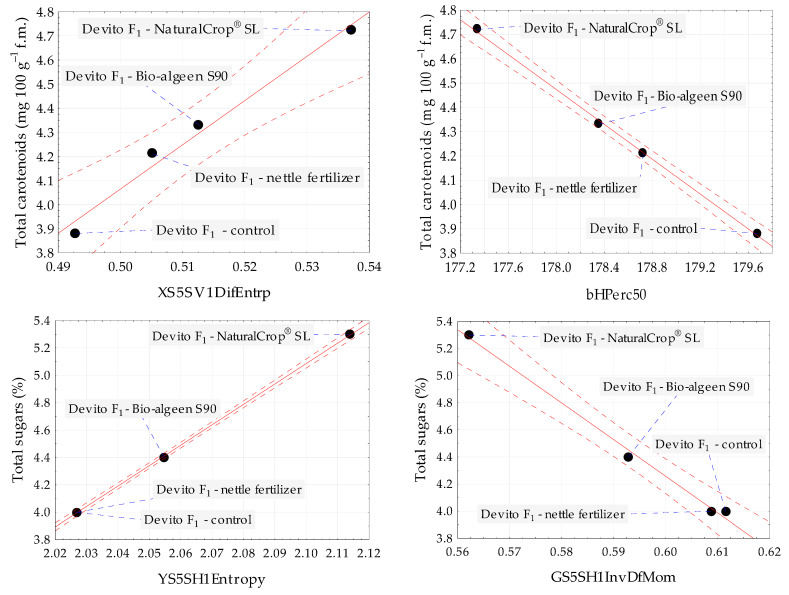
Scatter plots for total carotenoids and total sugars with selected image texture parameters of control and treated with NaturalCrop^®^, Bio-algeen S90, and nettle fertilizer yellow pepper Devito F_1_. Black dot—mean value; blue dashed line—line connecting the sample name to the mean value; red dashed line—confidence interval (95%); and solid red line—regression line.

**Figure 4 foods-12-02123-f004:**
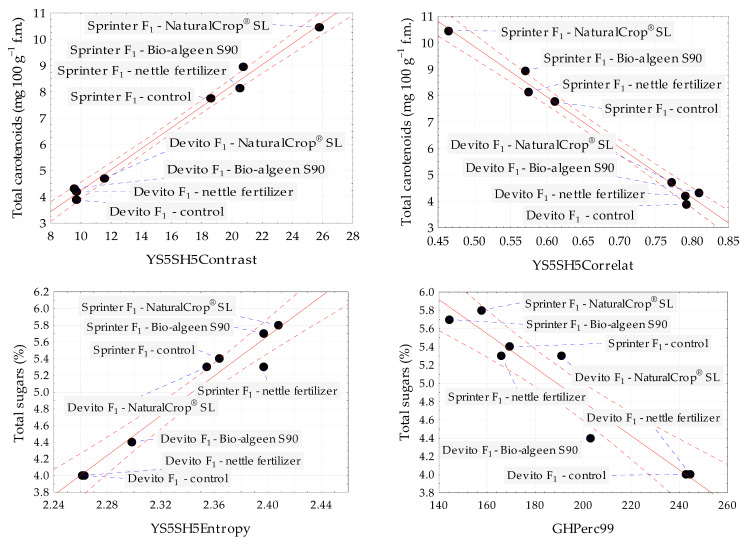
Scatter plots for total carotenoids and total sugars with selected image texture parameters of red pepper Sprinter F_1_ and yellow pepper Devito F_1_ belonging to control and samples treated with NaturalCrop^®^, Bio-algeen S90, and nettle fertilizer. Black dot—mean value; blue dashed line—line connecting the sample name to the mean value; red dashed line—confidence interval (95%); and solid red line—regression line.

**Table 1 foods-12-02123-t001:** Correlation coefficients (R) between α-carotene (mg 100 g^−1^ f.m.), β-carotene (mg 100 g^−1^ f.m.), total carotenoids (mg 100 g^−1^ f.m.), and total sugars (%) with selected image texture parameters for control and treated with NaturalCrop^®^, Bio-algeen S90, and nettle fertilizer red pepper Sprinter F_1_ samples; *p* < 0.05.

α-Carotene (mg 100 g^−1^ f.m.)		β-Carotene (mg 100 g^−1^ f.m.)		Total Carotenoids (mg 100 g^−1^ f.m.)		Total Sugars (%)	
Texture Parameter	Correlation Coefficient	Texture Parameter	Correlation Coefficient	Texture Parameter	Correlation Coefficient	Texture Parameter	Correlation Coefficient
BS5SN3Entropy	0.9999	BS5SN3Entropy	0.9994	aS5SH3DifEntrp	0.9999	RS5SH5SumOfSqs	0.9998
ZS4RZFraction	0.9996	aS5SV5Entropy	0.9991	GS5SV5Contrast	0.9998	XS5SH3SumOfSqs	0.9708
aS5SV5Entropy	0.9989	ZS4RZFraction	0.9985	XS5SZ3SumOfSqs	0.9933	GS4RVFraction	0.9685
GS5SN1Entropy	0.9964	GS5SV5Entropy	0.9984	ZS5SH3DifEntrp	0.9984		
YS5SN5DifEntrp	0.9847	YS5SN5DifEntrp	0.9763	RS5SN3Entropy	0.9984	aS5SN1SumVarnc	−0.9998
RS5SN5DifEntrp	0.9766	RS5SN5DifEntrp	0.9666	BS5SH3Entropy	0.9920	ZHDomn01	−0.9997
				YS5SH5DifEntrp	0.9825	XS5SV5SumEntrp	−0.9993
YS4RVLngREmph	−0.9999	aS5SV1AngScMom	−0.9998	bS5SN5Entropy	0.9507	YHDomn10	−0.9993
aS5SV3AngScMom	−0.9996	GS5SV1AngScMom	−0.9993	LS5SN5DifEntrp	0.9505	RHMaxm01	−0.9981
RS5SZ5SumAverg	−0.9975	YS4RVLngREmph	−0.9987			LHDomn10	−0.9979
GS5SV1AngScMom	−0.9972	RS5SZ5SumAverg	−0.9971	LS5SZ5SumVarnc	−0.9999	GS5SZ5SumVarnc	−0.9790
BS5SV5AngScMom	−0.9968	BS5SV5AngScMom	−0.9939	GS5SV3InvDfMom	−0.9999		
ZS5SV5InvDfMom	−0.9920	ZS5SV5InvDfMom	−0.9859	RS5SZ3InvDfMom	−0.9999		
		XS5SN5InvDfMom	−0.9631	bHPerc10	−0.9998		
				aS5SN1InvDfMom	−0.9996		
				BS4RZLngREmph	−0.9986		
				YS5SH3InvDfMom	−0.9909		
				ZS5SH5Correlat	−0.9847		
				XS5SZ5InvDfMom	−0.9724		

The first letter in the texture parameter name means the color channel *L*, *a*, *b*, *R*, *G*, *B*, *X*, *Y*, or *Z*; Entropy—entropy; Fraction—fraction of image in runs; DifEntrp—difference entropy; LngREmph —long run emphasis; AngScMom—angular second moment; SumAverg—sum average; InvDfMom—inverse difference moment; Contrast—contrast; SumOfSqs—sum of squares; SumVarnc—sum variance; Perc—percentile; Domn—dominant; SumEntrp—sum entropy; and Maxm—maximum of moments.

**Table 2 foods-12-02123-t002:** Correlation coefficients (R) between total carotenoids (mg 100 g^−1^ f.m.) and total sugars (%) with selected image texture parameters for control and treated with NaturalCrop^®^, Bio-algeen S90, and nettle fertilizer yellow pepper Devito F_1_ samples; *p* < 0.05.

Total Carotenoids (mg 100 g^−1^ f.m.)		Total Sugars (%)	
Texture Parameter	Correlation Coefficient	Texture Parameter	Correlation Coefficient
XS5SV1DifEntrp	0.9899	YS5SH1Entropy	0.9999
aSGNonZeros	0.9840	bS5SH3SumEntrp	0.9997
bATeta4	0.9783	GSGMean	0.9997
YATeta2	0.9716	LS5SZ5Contrast	0.9995
BS5SV1DifVarnc	0.9708	XS5SN3Contrast	0.9990
LS5SV3DifEntrp	0.9595	BS5SH1DifVarnc	0.9915
		aS5SZ5Correlat	0.9752
bHPerc50	−0.9993		
RSGKurtosis	−0.9948	GS5SH1InvDfMom	−0.9971
XS5SV1InvDfMom	−0.9909	XS5SZ5InvDfMom	−0.9894
GS4RHLngREmph	−0.9765	BS5SZ5SumVarnc	−0.9875
BS5SH5SumVarnc	−0.9696	aS5SN5DifVarnc	−0.9870
LATeta2	−0.9681	LS5SH1InvDfMom	−0.9808
		YS5SV1AngScMom	−0.9553

The first letter in the texture parameter name means the color channel *L*, *a*, *b*, *R*, *G*, *B*, *X*, *Y*, or *Z*; DifEntrp—difference entropy; GNonZeros—percentage of pixels with nonzero gradient; Teta4—parametr θ4; Teta2—parametr θ2; DifVarnc—difference variance; Perc—percentile; Kurtosis—histogram’s kurtosis; InvDfMom—inverse difference moment; LngREmph—long run emphasis; SumVarnc—sum variance; Entropy—entropy; SumEntrp—sum entropy; GMean—absolute gradient mean; Contrast—contrast; and Correlat—correlation.

**Table 3 foods-12-02123-t003:** Correlation coefficients (R) between total carotenoids (mg 100 g^−1^ f.m.) and total sugars (%) with selected image texture parameters for red pepper Sprinter F_1_ and yellow pepper Devito F_1_ belonging to control and samples treated with NaturalCrop^®^, Bio-algeen S90, and nettle fertilizer; *p* < 0.05.

Total Carotenoids (mg 100 g^−1^ f.m.)		Total Sugars (%)	
Texture Parameter	Correlation Coefficient	Texture Parameter	Correlation Coefficient
YS5SH5Contrast	0.9956	YS5SH5Entropy	0.9768
GS5SV5Contrast	0.9941	XS5SZ5Entropy	0.9562
LS5SN5Contrast	0.9883	LS5SH3Entropy	0.9461
aS5SH5Contrast	0.9832	GS5SN5Entropy	0.9443
XS5SZ5DifEntrp	0.9773	bS5SZ5SumEntrp	0.9375
bS5SV5SumEntrp	0.9415	aSGNonZeros	0.9309
ZSGVariance	0.8966		
BS5SZ3Contrast	0.8953	GHPerc99	−0.9583
		YS5SH1AngScMom	−0.9537
YS5SH5Correlat	−0.9935	LS5SN5InvDfMom	−0.9536
GS5SH5Correlat	−0.9921	XS5SH3AngScMom	−0.9494
LS5SH5Correlat	−0.9901	aHMaxm01	−0.9210
aS5SH1AngScMom	−0.9882	bHPerc99	−0.8803
XS4RNLngREmph	−0.9756	BS5SZ3Correlat	−0.8565
bHDomn01	−0.9663		
BS5SH3Correlat	−0.9202		
ZS5SH3Correlat	−0.9109		

The first letter in the texture parameter name means the color channel *L*, *a*, *b*, *R*, *G*, *B*, *X*, *Y*, or *Z*; Contrast—contrast; DifEntrp—difference entropy; SumEntrp—sum entropy; Variance—histogram’s variance; Correlat—correlation; AngScMom—angular second moment; LngREmph—long run emphasis; Domn—dominant; GNonZeros—percentage of pixels with nonzero gradient; Perc—percentile; InvDfMom—inverse difference moment; and Maxm—maximum of moments.

**Table 4 foods-12-02123-t004:** The regression equations for chemical properties based on image parameters and coefficients of determination for red pepper Sprinter F_1_ and yellow pepper Devito F_1_ belonging to control samples and samples treated with NaturalCrop^®^, Bio-algeen S90, and nettle fertilizer; *p* < 0.05.

Regression Equation	Coefficient of Determination (R^2^)
Sprinter F_1_	
α-carotene (mg 100 g^−1^ f.m.) = −21.28 + 8.4805 × BS5SN3Entropy	0.9998
α-carotene (mg 100 g^−1^ f.m.) = 1.1611 − 0.1012 × YS4RVLngREmph	0.9999
ß-carotene (mg 100 g^−1^ f.m.) = −23.35 + 9.4824 × BS5SN3Entropy	0.9988
ß-carotene (mg 100 g^−1^ f.m.) = 2.3051 − 114.3 × aS5SV1AngScMom	0.9996
Total carotenoids (mg 100 g^−1^ f.m.) = −34.78 + 43.407 × aS5SH3DifEntrp	0.9998
Total carotenoids (mg 100 g^−1^ f.m.) = 67.258 − 0.8091 × LS5SZ5SumVarnc	0.9998
Total sugars (%) = −4.350 + 0.49032 × RS5SH5SumOfSqs	0.9996
Total sugars (%) = 34.265 − 0.3261 × aS5SN1SumVarnc	0.9996
Devito F_1_	
Total carotenoids (mg 100 g^−1^ f.m.) = −5.122 + 18.372 × XS5SV1DifEntrp	0.9798
Total carotenoids (mg 100 g^−1^ f.m.) = 68.248 − 0.3583 × bHPerc50	0.9986
Total sugars (%) = −26.24 + 14.916 × YS5SH1Entropy	0.9998
Total sugars (%) = 20.481 − 27.04 × GS5SH1InvDfMom	0.9942
Sprinter F_1_ and Devito F_1_	
Total carotenoids (mg 100 g^−1^ f.m.) = 0.24275 + 0.39985 × YS5SH5Contrast	0.9912
Total carotenoids (mg 100 g^−1^ f.m.) = 19.407 − 19.10 × YS5SH5Correlat	0.9870
Total sugars (%) = −22.91 + 11.908 × YS5SH5Entropy	0.9536
Total sugars (%) = 8.5299 − 0.0187 × GHPerc99	0.9166

The first letter in the texture parameter name means the color channel *L*, *a*, *b*, *R*, *G*, *B*, *X*, *Y*, or *Z*; Entropy—entropy; LngREmph—long run emphasis; AngScMom—angular second moment; DifEntrp—difference entropy; SumVarnc—sum variance; SumOfSqs—sum of squares; Perc—percentile; InvDfMom—inverse difference moment; Contrast—contrast; and Correlat—correlation.

## Data Availability

The data presented in this study are available on request from the corresponding author.

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
