# Peer review of "The Estimation of Chemical Properties of Pepper Treated with Natural Fertilizers Based on Image Texture Parameters"

_foods, 2023, doi:10.3390/foods12112123_

Round 1
Reviewer 1 Report
The following steps have been carried out before review process of the manuscript (foods-2381404) entitled “The estimation of chemical properties of pepper treated with natural fertilizers based on image texture parameters”. The related keywords were searched in Web of Science. Although the presented study was an original study, it was considered that the results of the study were not appropriate because the plant materials used in the study were genetically heterozygout and fruit color due to the carotene content showed qualitative inheritance. Formation of fruit color in pepper plant like many other plant species is governed by major genes and is not affected by environmental conditions.
I have not suggested my comments and corrections on mn.
Author Response
Comment: The following steps have been carried out before review process of the manuscript (foods-2381404) entitled “The estimation of chemical properties of pepper treated with natural fertilizers based on image texture parameters”. The related keywords were searched in Web of Science. Although the presented study was an original study, it was considered that the results of the study were not appropriate because the plant materials used in the study were genetically heterozygout and fruit color due to the carotene content showed qualitative inheritance. Formation of fruit color in pepper plant like many other plant species is governed by major genes and is not affected by environmental conditions.
I have not suggested my comments and corrections on mn.
Response: Thank you very much for your comments. It has been rightly pointed out that the formation of fruit color in pepper plant is governed by major genes. The objective of this study was to estimate the chemical properties of pepper treated with natural fertilizers in a fast and inexpensive manner based on image texture parameters. The effect on color, which is known to depend on the variety and is genetically determined, was not examined. In this experiment, the effect of the preparations on the content of sugars and carotenoids was examined separately for each variety. It was also possible to determine regression models that are independent of the variety. Environmental conditions, such as natural fertilizers in our case, influenced the chemical properties of pepper fruits, while also affecting the structure of the flesh, which was visible in the textures of the image. The estimation of α-carotene, β-carotene, total carotenoids, and total sugars of peppers subjected to different organic fertilization using regression equations including features selected from a set of 1629 image textures from 9 color channels is a great novelty of the present study.
The image textural features considered in our study can be defined as a function of the spatial variation of the brightness intensity of the pixels. Images are characterized by repeated subpatterns of dispersion and distribution of pixel brightness, which represent the brightness, smoothness, roughness, size, directivity, and granulation of textures. Textures can carry important information about the structure of the examined physical objects. Quantitative analysis of texture parameters can provide diagnostic information about product quality. Texture analysis provides numerical data computed from the image of the object. Texture parameters can even specify the changes that are difficult to relate to changes that are perceived visually. In their digital image form, objects can be characterized by different textures even if they are composed of the same number of pixels and the same color histogram.
There are many references on this topic:
Armi, L.; Fekri-Ershad, S. Texture image analysis and texture classification methods—A review. Int. Online J. Image Process.Pattern Recogn. 2019, 2, 1–29.
Fernández, L.; Castillero, C.; Aguilera, J.M. An application of image analysis to dehydration of apple discs. J. Food Eng. 2005, 67, 185–193.
Strzelecki, M., Szczypiński, P., Materka, A., Klepaczko, A., 2013. A software tool for automatic classification and segmentation of 2D/3D medical images. Nucl. Instrum. Methods Phys. Res. 702, 137-140.
Ropelewska, E., 2020. The use of seed texture features for discriminating different cultivars of stored apples. J. Stored Prod. Res. 88, 1-7, 101668.
Ropelewska, E. The Application of Machine Learning for Cultivar Discrimination of Sweet Cherry Endocarp. Agriculture 2021, 11, 6.
Furthermore, we included the following information in the manuscript:
“Among the compounds that largely determine the nutritional value of pepper fruits are the sugars that give them their sweet taste. Their content is genetically determined and can be modified by agrotechnical and climatic conditions [11-14].”
“The pepper fruit cultivars differ in color, size, shape, and flavor. The color and shape of pepper fruit at the stage of consumer maturity depend on the presence of a gene system. Pepper fruits synthesize and accumulate a variety of carotenoids in tissues with a wide range of content. Even within the same cultivar, the composition and number of carotenoids can vary considerably depending on maturity stage, growing region, and growing conditions [20-23]. The first level of regulation of carotenoid biosynthesis in pepper fruit is the transcriptional control of gene biosynthesis.”
“Flesh and skin texture, juiciness and postharvest shelf life are determined by both genetic traits and growing conditions [28].”
Reviewer 2 Report
This manuscript describes the use of photographic methods to predict the compositional traits (sugar and carotenoids contents) of bell peppers. The interest in such a study would be enhanced if the methods used to analyse these fruits were better described and if the software analysis done to interpret the image parameters was explained. Also, the conclusions point to the image analysis method as a novel route to estimate chemical properties of pepper fruits, but no such attempt was made by the authors, say on a commercial fruit, which is a pity.
I detail below some of my specific concerns that, once addressed, would in my view improve the manuscript:
- The image analysis software and the 9 colour channels mentioned throughout the text should be explicitly described at least once in the introduction. Lines 105-106 where the method is mentioned briefly does not accurately present the method and should be developed.
- Line 110-119 There is an intrinsic difficulty in drawing statistically relevant information on using an experimental design based on only 4 variables per pepper cultivar (5 groups are needed to adequately estimate a linear equation) Please explain your decision to restrict your study on 4 cultivation protocols.
- Describe adequately the fertilisers used (some commercial, others like nettle are not) so that readers can better understand the study.
- An image (or better still a series of images treated under each colour parameter) could be added to the manuscript or in supplementary materials, to illustrate the image acquisition and processing method. Also, an estimate of variability (e.g. %RSD) of the 181 textures for each of the 9 colour channels could be included here. Readers do not have access to this important information either in the materials and methods section neither in the results section.
- Section on chemical properties. Please expand the description of the HPLC method for carotenoid determination (gradient is mentioned but not described, injection volume is not given). Method for the sugar determination is referenced to a Polish standard unavailable outside of Poland. Please give full details of this HPLC method or cite a more accessible reference. Also, please give estimates of variability (e.g. %RSD) of the chemical contents determined by these HPLC methods. How many replicate analyses were made on each cultivar and each growing protocol?
Tables 1-4: the image texture parameters (e.g. BS5SN3Entropy etc. ) are never described and are as such meaningless to the reader. Please provide some explanation.
Figures 1-3: Data presented in forms of data-points without error bars (either on chemical composition nor on image parameter) I understand that these data points are averages of multi data points yet, no standard deviation is given on any measurement. These scatter plot figures also appear to present lines of confidence intervals ( CI 95%??) yet are not described and not expanded upon. Authors instead use a very reducing R2 parameter to show goodness of fit.
Table 4 presents line equations with only R2 as estimates of goodness of fit. It is obvious that better fits (as defined narrowly by R2) will be obtained with lower sloped equations, yet these image parameters will be less able to discriminate between different pepper composition. Please comment in the discussion instead of exclusively comparing image parameters based on R2.
Before being able to conclude that the image analysis methods presented could be used to estimate chemical composition, some of these experiments should be done on, for example, a commercial pepper to establish the limits of validity of the method.
Author Response
This manuscript describes the use of photographic methods to predict the compositional traits (sugar and carotenoids contents) of bell peppers. The interest in such a study would be enhanced if the methods used to analyse these fruits were better described and if the software analysis done to interpret the image parameters was explained. Also, the conclusions point to the image analysis method as a novel route to estimate chemical properties of pepper fruits, but no such attempt was made by the authors, say on a commercial fruit, which is a pity.
Response: Thank you very much for your opinion. The methods are described in more detail according to the comments below.
For the experiment, cultivars of pepper that are commonly used in organic greenhouse cultivation were selected. The used pepper cultivars are commercially available to consumers. However, the experiment involving an approach using selected natural fertilizers was novel. Therefore, the research material was grown in our own greenhouse. It would not be possible to buy fruits treated with these fertilizers in the store.
I detail below some of my specific concerns that, once addressed, would in my view improve the manuscript:
Comment: - The image analysis software and the 9 colour channels mentioned throughout the text should be explicitly described at least once in the introduction. Lines 105-106 where the method is mentioned briefly does not accurately present the method and should be developed.
Response: It has been described in more detail as follows:
“Image textures can be analyzed using specialized software packages, for example, MaZda, which enables quantitative analysis, including texture parameters computation, as well as texture selection and extraction, data classification and visualization, and image segmentation tools. MaZda provides options for the transformation of color images to grayscale and conversions to individual color channels (components) [31-33]. Several channels are most commonly used. There are color channels R (red), G (green), B (blue) from the RGB color space, L (lightness component from black to white), a (red for positive and green for negative values), b (yellow for positive and blue for negative values) from the Lab color space, and X (component with color information), Y (lightness), Z (component with color information) from the XYZ color space [29,34].”
- Ropelewska, E. The Application of Machine Learning for Cultivar Discrimination of Sweet Cherry Endocarp. Agriculture 2021, 11, 6.
- Strzelecki, M.; Szczypinski, P.; Materka, A.; Klepaczko, A. A software tool for automatic classification and segmentation of 2D/3D medical images. Nuclear Instruments and Methods in Physics Research Section A: Accelerators, Spectrometers, Detectors and Associated Equipment 2013, 702, 137-140.
- Szczypiński, P. M., Strzelecki, M.; Materka, A.; Klepaczko, A. MaZda - A software package for image texture analysis. Computer Methods and Programs in Biomedicine 2009, 94(1), 66-76.
- Szczypiński, P.M.; Strzelecki, M.; Materka, A. Mazda-a software for texture analysis, 2007 international symposium on information technology convergence (ISITC 2007). IEEE, 2007, pp. 245-249.
- Ibraheem, N.A.; Hasan, M.M.; Khan, R.Z.; Mishra, P.K. Understanding Color Models: A Review. ARPN J. Sci. Technol. 2012, 2, 265–275.
Comment: - Line 110-119 There is an intrinsic difficulty in drawing statistically relevant information on using an experimental design based on only 4 variables per pepper cultivar (5 groups are needed to adequately estimate a linear equation) Please explain your decision to restrict your study on 4 cultivation protocols.
Response: For this experiment, in addition to control samples, natural preparations were selected: with a killing and nutritional effect (nettle extract) and fertilizers approved in organic farming: Bio-algeen S90 (organic fertilizer from sea algae), NaturalCrop® SL (growth and development stimulator), which seems sufficient to compare their effects on 2 cultivars of peppers, since each of them shows a different direction of action. Figure 4 presents all samples belonging to 2 cultivars and 4 groups together, so linear relationships were determined for 8 groups.
The information on experimental design and preparations has been supplemented as follows:
“2.1. Experimental design
The experimental material consisted of pepper growing in greenhouses at the Laboratory of Cultivation of Vegetable and Edible Mushrooms of the National Institute of Horticultural Research in Skierniewice in Poland. Peppers belonged to two cultivars of Sprinter F1 characterized by red fruit and Devito F1 with yellow fruit. The experiment was carried out between March and October 2021. During the experiment, daily climatological data were collected inside the greenhouse, using a thermo-hygrometer. Organic pepper seeds were used and potted seedlings were planted in the soil in an ecological greenhouse under the principles of organic farming. The experiment was set up in the soil with a pH of 6.0. Before planting, the soil was fertilized with organic plant compost at a dose of 300 kg per 100 m2, obtaining the soil fertility at the level (mg dm−3) of N-230, P-310, K-400, and Mg-350. The seedlings were planted on April 20, in a row-row system, at a spacing of 35 cm × 60 cm (3.6 plants per 1 m2). Plants were grown in a 4-shoot system. Pepper fertigation was carried out with the use of a drip system and fertilizers approved for organic cultivation depending on the current humidity and nutritional requirements of the pepper in particular growth stages [35]. The fruit was harvested at the stage of colorful ripeness.
2.2. Preparations
In research natural preparation and fertilizers approved for organic farming was performed such as: nettle fertilizer (NE/427/2018) (as a substance with biocidal and nutritional effect) based on N-NO3 about 0.5-0.7 mg·l-1, N-NH4 >100 mg·l-1, P about 20 mg·l-1, K-about 650 mg·l-1, F - 0.2 mg·l-1, pH 6.3-6.7 (20% solution was used); Bio-algeen S90 (organic fertilizer) containing 90 groups of chemicals compounds including: amino acids, vitamins, alginic acid and other unexplored active components of marine algae, N - 0.02%; P - 0.006%; K - 0.096%; Ca - 0.31%; MG - 0.021% and B - 16 mg·kg-1; Fe - 6.3 mg·kg-1; Cu - 0.2 mg·kg-1; Mn - 0.6 mg·kg-1; Zn -1.0 mg·kg-1 (0.3% solution was used); NaturalCrop® SL (growth and development stimulator) - enzymatic concentrate of peptides and 16 L-amino acids most important for plants (GLY, GLU, HIS, LYS) of plant origin >50% (605 g·l-1), including free amino acids 0.2% (242g·l-1), amino nitrogen 9% (109 g·l-1), organic carbon 24.5% (296.5 g·l-1) (0.3% solution was used) and the control (unfertilized plots).”
Comment: - Describe adequately the fertilisers used (some commercial, others like nettle are not) so that readers can better understand the study.
Response: The following natural preparations were used: with a killing and nutritional effect (nettle extract) and fertilizers approved in organic farming: Bio-algeen S90 (organic fertilizer from sea algae), NaturalCrop® SL (growth and development stimulator). The description was supplemented as follows:
“2.2. Preparations
In research natural preparation and fertilizers approved for organic farming was performed such as: nettle fertilizer (NE/427/2018) (as a substance with biocidal and nutritional effect) based on N-NO3 about 0.5-0.7 mg·l-1, N-NH4 >100 mg·l-1, P about 20 mg·l-1, K-about 650 mg·l-1, F - 0.2 mg·l-1, pH 6.3-6.7 (20% solution was used); Bio-algeen S90 (organic fertilizer) containing 90 groups of chemicals compounds including: amino acids, vitamins, alginic acid and other unexplored active components of marine algae, N - 0.02%; P - 0.006%; K - 0.096%; Ca - 0.31%; MG - 0.021% and B - 16 mg·kg-1; Fe - 6.3 mg·kg-1; Cu - 0.2 mg·kg-1; Mn - 0.6 mg·kg-1; Zn -1.0 mg·kg-1 (0.3% solution was used); NaturalCrop® SL (growth and development stimulator) - enzymatic concentrate of peptides and 16 L-amino acids most important for plants (GLY, GLU, HIS, LYS) of plant origin >50% (605 g·l-1), including free amino acids 0.2% (242g·l-1), amino nitrogen 9% (109 g·l-1), organic carbon 24.5% (296.5 g·l-1) (0.3% solution was used) and the control (unfertilized plots).”
Comment: - An image (or better still a series of images treated under each colour parameter) could be added to the manuscript or in supplementary materials, to illustrate the image acquisition and processing method. Also, an estimate of variability (e.g. %RSD) of the 181 textures for each of the 9 colour channels could be included here. Readers do not have access to this important information either in the materials and methods section neither in the results section.
Response: Figure 1 with some exemplary original color images of pepper slices belonging to two cultivars and images in 9 color channels has been added.
It would be very difficult to indicate %RSD or another variability parameter for 181 textures for each of the 9 color channels. Each texture has different values and different variability. However, the data are available on request from the corresponding author so readers would have access to this information.
Comment: - Section on chemical properties. Please expand the description of the HPLC method for carotenoid determination (gradient is mentioned but not described, injection volume is not given). Method for the sugar determination is referenced to a Polish standard unavailable outside of Poland. Please give full details of this HPLC method or cite a more accessible reference. Also, please give estimates of variability (e.g. %RSD) of the chemical contents determined by these HPLC methods. How many replicate analyses were made on each cultivar and each growing protocol?
Response: Subsection 2.4. Chemical properties has been expanded as follows:
“2.4. Chemical properties
The samples for chemical analysis were prepared from 10 pepper fruits from each repetition. Pepper fruits were cut into quarters, frozen in liquid nitrogen, crushed in a Blixer homogenizer and stored at −20 °C.
2.4.1. HPLC analysis of sugars
An HPLC analysis of total sugars (sucrose, glucose and fructose) was determined by high-performance liquid chromatography (Agilent 1200 HPLC system, equipped with a differential refractometric detector), using Aminex HPX-87C (300 mm x 7.5 mm) with a precolumn according to European Standard EN 12630. The isocratic flow was 0.6 mL·min-1, column temperature was 80 °C , and mobile phase-edetate calcium disodium (Ca-EDTA). The samples were dissolved in redistilled water, homogenized and purified on a Waters SepPak PLUS C18 filter. The sugars were quantified by calibration curve for sucrose, glucose and fructose, and the results were expressed as mg·100 g-1 fresh mass (FM).
2.4.2. Carotenoid extraction
Carotenoid content was determined by the method according to Bohoyo-Gil [36]. The sample was homogenized in the extraction solution (hexane:acetone 6:4) and filtered through a Büchner funnel under reduced pressure. Next, the extract was transferred to a separating funnel and shaken with the addition of water. After phase separation, the water-acetone phase was discarded. The acetone rinsing operation was repeated until the lower phase was free of acetone and the upper hexane phase containing carotenoids was filtered through a filter paper containing anhydrous sodium sulphate into an evaporation flask. Hexane was evaporated to dryness in a vacuum evaporator at 40°C, the dry residue was quantitatively transferred to a 25 ml flask with a solution of acetonitrile:methanol:ethyl acetate 55:25:20 + 0.1% BHT + 1 ml TEA and 4 ml of hexane. The flask extract was filtered with a 45 µm PTFE filter into an amber bottle and analyzed by HPLC.
2.4.3. HPLC analysis of carotenoids
An HPLC analysis of total carotenoids in pepper samples was determined by high-performance liquid chromatography (HPLC), using a Kinetex C-18 column (250 mm x 4.6 mm; 5 µm) on an Agilent 1200 HPLC system equipped with a DAD detector. The elution conditions were as follows: 0.7 ml min-1, temperature 28 °C, wavelength 450 nm and 472 nm, mobile phase: acetonitrile (A), methanol + 1 ml TEA + 1 g BHT (B), ethyl acetate (C), in gradient flow: 0-10 min, 95% A, 2% B, 3% C; 10-25 min, 55% A, 20% B, 25% C; 25-35 min, 95% A, 2% B, 3% C. The calculations were quantified by calibration with the standards of β-carotene and lycopene (Sigma-Aldrich, Germany). The carotenoid content was expressed in mg·100g-1 FM.”
Comment: Tables 1-4: the image texture parameters (e.g. BS5SN3Entropy etc. ) are never described and are as such meaningless to the reader. Please provide some explanation.
Response: It has been explained in the footnotes below each Table.
Comment: Figures 1-3: Data presented in forms of data-points without error bars (either on chemical composition nor on image parameter) I understand that these data points are averages of multi data points yet, no standard deviation is given on any measurement. These scatter plot figures also appear to present lines of confidence intervals ( CI 95%??) yet are not described and not expanded upon. Authors instead use a very reducing R2 parameter to show goodness of fit.
Response: The data presented in Figures have been better explained as follows:
“Black dot — mean value; blue dashed line — line connecting the sample name to the mean value; red dashed line — confidence interval (95%); solid red line — regression line.”
Comment: Table 4 presents line equations with only R2 as estimates of goodness of fit. It is obvious that better fits (as defined narrowly by R2) will be obtained with lower sloped equations, yet these image parameters will be less able to discriminate between different pepper composition. Please comment in the discussion instead of exclusively comparing image parameters based on R2.
Response: The discussion has been supplemented as you suggested.
Comment: Before being able to conclude that the image analysis methods presented could be used to estimate chemical composition, some of these experiments should be done on, for example, a commercial pepper to establish the limits of validity of the method.
Response: For the experiment, cultivars of pepper that are commonly used in organic greenhouse cultivation were selected. The used pepper cultivars are commercially available to consumers. However, the experiment involving an approach using selected natural fertilizers was novel. Therefore, the research material was grown in our own greenhouse. It would not be possible to buy fruits treated with these fertilizers in the store.
Reviewer 3 Report
The authors have presented an interesting paper which evaluated the estimation of chemical properties of pepper treated with natural fertilizers based on image texture parameters. The topic of this manuscript is very interesting, since in the last decades the excessive chemical fertilization provoked many damages in the environment. Following, I have included some comments aimed to enhance the paper:
1. I suggest to the authors to add a new section detailing the state of the art. In this section, authors have to describe the relevant related work in which explain the use of amended in soils and their effects, authors have to identify the innovation of their study with other existing, and to cite also some results of the efficacy of the use of organic residues.
2. Can the authors include at the end of the introduction, more details of the objectives of their study, sine they are comparing different organic residues.
3. Consider extending the conclusions and adding a Future works paragraph. The summary and Conclusions, it is better to combine them in only section of conclusions.
Finally, we consider this work very interesting, explaining the changes in chemical properties of pepper with the application of natural fertilizers. The chemical changes on pepper can help extremely (in other regions with similar soils and climate) in the preservation of environment, the reduction of pollution and the protection of soil.
Author Response
The authors have presented an interesting paper which evaluated the estimation of chemical properties of pepper treated with natural fertilizers based on image texture parameters. The topic of this manuscript is very interesting, since in the last decades the excessive chemical fertilization provoked many damages in the environment. Following, I have included some comments aimed to enhance the paper:
Comment 1. I suggest to the authors to add a new section detailing the state of the art. In this section, authors have to describe the relevant related work in which explain the use of amended in soils and their effects, authors have to identify the innovation of their study with other existing, and to cite also some results of the efficacy of the use of organic residues.
Response: The innovation has been indicated as follows:
“Own results presented in this paper were a great novelty compared to the available literature, as included the determination of correlations and models for predicting the content of chemical properties based on selected parameters from a very large set of over 1600 textures from 9 color channels. It allowed for the selection of textures providing correlation coefficients of up to 0.9999 and -0.9999, as well as very high coefficients of determination reaching 0.9999. The successful results may prompt the use of the developed procedure for other species and varieties of fruit and vegetables and testing the approach for other chemical parameters.”
Regarding the state of the art, we wrote another paper on the influence of natural preparations on the chemical composition, classification performance, and sensory quality of pepper fruit in organic greenhouse cultivation, which is currently under review. That paper included the relevant related work on organic fertilizers and their effects. We would not like to repeat it in this article. However, if the Reviewer still recommends adding a new section detailing the state of the art, of course, we will do it.
Comment 2. Can the authors include at the end of the introduction, more details of the objectives of their study, sine they are comparing different organic residues.
Response: It has been corrected to be more detailed as follows:
“The objective of this study was to estimate the chemical properties of peppers be-longing to two cultivars treated with natural fertilizers in a fast and inexpensive manner based on image texture parameters. A great novelty of the present study is related to the estimation of α-carotene, β-carotene, total carotenoids, and total sugars of untreated red pepper Sprinter F1 and yellow pepper Devito F1 and samples subjected to different organic fertilization, such as NaturalCrop® SL, Bio-algeen S90, and nettle fertilizer. The innovative regression equations including features selected from a set of 1629 image textures from 9 color channels were determined. The developed procedure could enable the selection of fruit with the desired properties based on the image features without the need to perform more labor-intensive and expensive measurements.”
Comment 3. Consider extending the conclusions and adding a Future works paragraph. The summary and Conclusions, it is better to combine them in only section of conclusions.
Response: The section Conclusions has been corrected and supplemented with future works.
The following sentences on future works have been added:
“Due to the achievement of promising results, research aimed at estimating chemical properties based on image features can be continued and further research directions can be set. Further research may include more pepper cultivars, as well as other species of fruit and vegetables. Furthermore, more natural fertilizers can be used to provide more groups in an experimental design. The developed procedure can be of great practical importance to estimate the chemical properties of samples treated with various fertilizers based on selected image texture parameters.”
Comment: Finally, we consider this work very interesting, explaining the changes in chemical properties of pepper with the application of natural fertilizers. The chemical changes on pepper can help extremely (in other regions with similar soils and climate) in the preservation of environment, the reduction of pollution and the protection of soil.
Response: Thank you very much for this comment.
Reviewer 4 Report
In the present study, Authors focused on the correlation between α-carotene, β-carotene, total carotenoids, and total sugars of red and yellow peppers subjected to different organic fertilization with different color channels in order to provide information regarding the possibility of estimate chemical properties of pepper fruit based on selected image textures.
This topic is of interest in agriculture sector for the development of technologies capable to help farmers predict the qualitative aspects of agricultural products before harvesting.
The article is well structured although some information should be improved, especially in Material and Methods section. Specifically in:
[111] 2.1 Experimental design: if possible, please add soil characteristic (texture, ph etc..)
[118] add the dose of fertilizers applied.
[122] please report how slices were produced (ei., by hands or using a machinery)
[172] Results and discussion?
Within conclusion section I would recommend Authors to add few sentences on future application of their findings in order to strengthen the manuscript
Good
Author Response
In the present study, Authors focused on the correlation between α-carotene, β-carotene, total carotenoids, and total sugars of red and yellow peppers subjected to different organic fertilization with different color channels in order to provide information regarding the possibility of estimate chemical properties of pepper fruit based on selected image textures.
This topic is of interest in agriculture sector for the development of technologies capable to help farmers predict the qualitative aspects of agricultural products before harvesting.
The article is well structured although some information should be improved, especially in Material and Methods section. Specifically in:
Comment: [111] 2.1 Experimental design: if possible, please add soil characteristic (texture, ph etc..)
Response: It has been supplemented as follows:
“2.1. Experimental design
The experimental material consisted of pepper growing in greenhouses at the Laboratory of Cultivation of Vegetable and Edible Mushrooms of the National Institute of Horticultural Research in Skierniewice in Poland. Peppers belonged to two cultivars of Sprinter F1 characterized by red fruit and Devito F1 with yellow fruit. The experiment was carried out between March and October 2021. During the experiment, daily climatological data were collected inside the greenhouse, using a thermo-hygrometer. Organic pepper seeds were used and potted seedlings were planted in the soil in an ecological greenhouse under the principles of organic farming. The experiment was set up in the soil with a pH of 6.0. Before planting, the soil was fertilized with organic plant compost at a dose of 300 kg per 100 m2, obtaining the soil fertility at the level (mg dm−3) of N-230, P-310, K-400, and Mg-350. The seedlings were planted on April 20, in a row-row system, at a spacing of 35 cm × 60 cm (3.6 plants per 1 m2). Plants were grown in a 4-shoot system. Pepper fertigation was carried out with the use of a drip system and fertilizers approved for organic cultivation depending on the current humidity and nutritional requirements of the pepper in particular growth stages [35]. The fruit was harvested at the stage of colorful ripeness.”
Comment: [118] add the dose of fertilizers applied.
Response: It has been corrected as follows:
“Before planting, the soil was fertilized with organic plant compost at a dose of 300 kg per 100 m2, obtaining the soil fertility at the level (mg dm−3) of N-230, P-310, K-400, and Mg-350.”
“2.2. Preparations
In research natural preparation and fertilizers approved for organic farming was performed such as: nettle fertilizer (NE/427/2018) (as a substance with biocidal and nutritional effect) based on N-NO3 about 0.5-0.7 mg·l-1, N-NH4 >100 mg·l-1, P about 20 mg·l-1, K-about 650 mg·l-1, F - 0.2 mg·l-1, pH 6.3-6.7 (20% solution was used); Bio-algeen S90 (organic fertilizer) containing 90 groups of chemicals compounds including: amino acids, vitamins, alginic acid and other unexplored active components of marine algae, N - 0.02%; P - 0.006%; K - 0.096%; Ca - 0.31%; MG - 0.021% and B - 16 mg·kg-1; Fe - 6.3 mg·kg-1; Cu - 0.2 mg·kg-1; Mn - 0.6 mg·kg-1; Zn -1.0 mg·kg-1 (0.3% solution was used); NaturalCrop® SL (growth and development stimulator) - enzymatic concentrate of peptides and 16 L-amino acids most important for plants (GLY, GLU, HIS, LYS) of plant origin >50% (605 g·l-1), including free amino acids 0.2% (242g·l-1), amino nitrogen 9% (109 g·l-1), organic carbon 24.5% (296.5 g·l-1) (0.3% solution was used) and the control (unfertilized plots).”
Comment: [122] please report how slices were produced (ei., by hands or using a machinery)
Response: It has been corrected as follows:
“The images of pepper slices (cross-sections) with a thickness of 10 mm obtained using a slicing machine were considered.”
Comment: [172] Results and discussion?
Response: It has been corrected.
Comment: Within conclusion section I would recommend Authors to add few sentences on future application of their findings in order to strengthen the manuscript
Response: It has been supplemented as follows:
“Due to the achievement of promising results, research aimed at estimating chemical properties based on image features can be continued and further research directions can be set. Further research may include more pepper cultivars, as well as other species of fruit and vegetables. Furthermore, more natural fertilizers can be used to provide more groups in an experimental design. The developed procedure can be of great practical importance to estimate the chemical properties of samples treated with various fertilizers based on selected image texture parameters.”
Round 2
Reviewer 1 Report
Dear Author,
I have read your responses.
Best regards,
Reviewer 2 Report
Thanks to the authors for having taken comments into account and expanded on the details of analysis and presentation.